# 'Just Play' (JP) - creative arts therapies-based dyadic intervention for children with intellectual disability and their mothers: Study protocol for a mixed-methods randomized controlled trial

Rita Abramov [ID]°*, Amitai Stern [ID]°, Rinat Feniger-Schaal, Cochavit Elefant, Limor Goldner, Tal-Chen Rabinowitch

School of Creative Arts Therapies, Faculty of Social Welfare and Health Sciences, University of Haifa, Haifa, Israel

☉ These authors contributed equally to this work.
* rita.abramov@gmail.com

## Abstract

### Background

Play is a fundamental aspect of children's development, fostering their cognitive, social, and emotional growth. However, children with intellectual disability (ID) experience limitations in their ability to engage in play, which can impact their relationships with caregivers. Parents of children with ID face heightened stress and reduced playfulness in their interactions with their children. Existing early interventions have primarily focused on behavioral parent training programs to manage challenging child behaviors, lacking a direct and positive approach to enhancing parent-child play interactions and relationships.

### Objective

This study aims to develop, implement, and evaluate 'Just Play'(JP), a creative arts therapies-based dyadic early intervention to enhance the quality of mother-child interactions through positive playful engagement.

### Methods

A mixed-methods randomized controlled trial will be conducted with 60 mother-child dyads (children aged 3–7 with a prior ID diagnosis). Participants will be randomly assigned to either the JP dyadic intervention or to psycho-educational parent counseling control group. Quantitative data will include measures of emotional availability, mother and child playfulness, and the level of interpersonal synchrony between mothers and their children at baseline and post-intervention. Additionally, qualitative interviews with a subset of mothers in the JP group will be conducted at post-intervention

---

**Data availability statement:** The authors plan to make the datasets generated and analyzed during the study along with study materials (e.g., intervention protocols, questionnaires) available by depositing them in the Open Science Framework public repository (https://doi.org/10.17605/OSF.IO/YMCNH).

**Funding:** This study received funding through a competitive peer-reviewed process on the topic of Children with Cognitive Challenges, from an anonymous private donation to the Faculty of Welfare and Health Sciences at the University of Haifa. The external funder had no role in study design, data collection and analysis, decision to publish, or preparation of the manuscript but is updated in yearly reports provided by the investigators. AS received Shalem Fund grant no. 890-709 for this study.

**Competing interests:** We declare that none of the authors have potential competing interests. The authors are not aware of any competing interests regarding the donor.

and two-month follow-up to explore their experiences and identify the best practices of the intervention.

## Results & Implications

This study will provide empirical evidence on the effectiveness of JP dyadic intervention in strengthening positive mother-child relationships in families of children with ID. Findings will inform future early intervention strategies, contributing to evidence-based guidelines for practitioners working with children with ID while emphasizing the unique contribution of creative arts therapies to this population.

## Trial registration

ClinicalTrials.gov, registered August 6, 2024 (NCT06541782 || https://clinicaltrials.gov/study/NCT06541782).

## Background

Play is a universal activity through which children acquire early knowledge of the world and promote their cognitive, linguistic, social, and emotional development [1,2]. Contemporary developmental theories conceptualize play as an intrinsically motivated, voluntary, and pleasurable activity that is driven by joy and engagement rather than external goals [3–5]. Major theorists, including Piaget, Vygotsky, and subsequent developmental scholars, have emphasized play as a primary context for learning, symbolic representation, self-regulation, and social understanding across childhood [1–4].

The development of children's play parallels the progression of their general development [4]. The first stage of sensorimotor play, appears in early infancy, and involves sensory exploration and bodily motor coordination and movement [1]. This is followed by object play, in which children use objects functionally and symbolically, and subsequently by pretend play and social play, developmental milestones associated with abstract thinking, flexible cognition, and advanced language skills [1,5]. Although play skills typically increase in complexity with development, play trajectories are non-linear and shaped by individual differences and environmental interactions [6]. Furthermore, while most children acquire play skills through observations and responsive interactions with caregivers, children with disabilities may need intentional interventions to develop certain play abilities [7].

Children with intellectual disability (ID) constitute a heterogeneous group characterized by significant limitations in intellectual functioning and adaptive behavior with onset in the developmental period. These limitations affect multiple developmental domains including language, motor skills, communication, and perception, and substantially constrain opportunities for play, exploration, and social engagement [6,7]. Research on play development indicates that children with ID exhibit less sophisticated play patterns compared to their typically developing (TD) peers, often resembling those observed in much younger children. Specifically, children with ID

demonstrate a predominance of sensorimotor and physical forms of play [8,9]. Studies examining play preferences among children aged 3–7 years further reveal that children with ID show stronger preferences for sensorimotor rough-and-tumble play and object exploration, alongside lower preferences for symbolic and pretend play relative to TD children [10]. Within sensorimotor play, limited motor coordination frequently constrains performance, leading children with ID to favor physical activities that require minimal precision and coordination [9]. These motor challenges, along with other difficulties, influence their object play, which tends to be repetitive and less elaborated, characterized by non-exploratory behaviors such as mouthing, pounding, or squeezing objects [9]. Symbolic play emerges more slowly in this population, resulting in pretend play that is less imaginative, elaborated and flexible compared to TD children [9,11,12].

Children with ID experience delays in the development of their social skills [11], and this is reflected in their social play: Research shows children with ID have difficulties in maintaining cooperative play with others, initiating or joining play, taking turns, or sharing toys [13,14]. Interestingly, familiar play activities were found to facilitate higher levels of social interaction in children with ID than more demanding novel tasks [15]. These deficiencies profoundly influence the social development of children with ID and may lead to long-lasting emotional and behavioral issues that affect not only the children but also their parents, affecting the early stages of their relationships [12,16].

Rearing a child with ID can be demanding, with research suggesting that parents of children with ID are at a high risk of encountering elevated stress levels, social isolation and attenuated psychological well-being [12]. The parenting experience can be further challenged by the behavioral and emotional difficulties that often emerge in children with ID. Indeed, the risk of developing behavior disorders among children with developmental disabilities, such as ID, is three to four times higher than among TD children, placing them at a heightened risk for the emergence of comorbid psychopathology [17,18]. These challenges between parents and children with ID manifest themselves in lower levels of parental emotional availability (EA) [19] and parent-child interpersonal synchrony (IPS) [20], two constructs that have been extensively researched and shown to be central to the parent-child relationship and to child development in general [21–23].

EA refers to the dyad's capacity to establish and maintain a healthy relationship, focusing on the parent's ability to understand and respond to the child's needs [24]. Emotional transactions between parents and children are evaluated according to six categories: parental sensitivity, parental structuring, parental non-intrusiveness, parental non-hostility, child responsiveness, and child involvement. High levels of parental EA are associated with secure attachment, positive parent-child relationships, and favorable child outcomes, including emotional regulation and social development [24]. Parent-child IPS refers to the dynamic and reciprocal adaptation of movements and behaviors, both verbal and non-verbal, between the parent and child, such as postures, gestures, breathing patterns, vocalizations, and facial expressions [22]. IPS in turn lays the foundation for the child's ability to form intimate relationships in the future and thereby plays a pivotal role in developing self-regulation, positive self-esteem, empathy, and peer competence [20,23]. Parents and children with ID tend to share fewer moments of joint attention and engagement than TD children, displaying fewer patterns of symmetrical behavior and lower parent-child synchrony [25]. Studies also suggest that mothers of children with ID tend to exhibit fewer sensitive responses and engage in more intrusive and directive behaviors, often relying on commands and physical control [26,27]. Thus, recent research on children with ID emphasizes the need for early interventions that focus on fostering positive parent-child relationships, given the unique challenges that parents of children with ID may encounter and the critical role that the parent-child relationship plays on shaping the child's developmental and emotional outcomes [17,28,29].

While intervention programs for children with ID have traditionally focused on addressing their developmental challenges, recent studies have shown more early intervention programs that target parenting and dyadic communication skills in this population [29]. Among the most salient programs for behavioral parent training is the Incredible Years Parent Training, adapted for use with parents of preschool children with mixed developmental delays (IYPT-DD) [30]. The program is based on 12 weekly group sessions for parents and employs discussion, video modeling, role-playing, and psycho-education on raising a child with developmental delays, with a focus on fostering positive parent-child interactions.

                                                 

The IYPT-DD has been shown to be efficacious in reducing negative parent-child interactions and child behavior problems, providing evidence that parenting processes can improve problematic behaviors in young children with developmental delays [31]. Similar behavioral parent training programs that focus on reducing child undesired behaviors include the Stepping Stones Triple P [32] and Parent-Child Interaction Therapy [33], with the former being a parent training program performed in a group format and the latter being performed in an individual format and adapted for parents of children with both ID and oppositional defiant disorder [32–34]. Both of these programs demonstrated success in mitigating children's challenging behaviors and negative parenting behaviors, drawing on principles guided by applied behavior analysis and providing parents with behavior management strategies that reinforce children's positive conduct [34]. While these interventions offer evidence that constructive interaction patterns between parents and children with ID can be taught and favorably affect behavioral outcomes, they generally do not emphasize fostering positive and playful parent-child interactions, but rather focus on providing specific strategies for parents to deal with problematic behaviors of their children with ID. Instead of primarily addressing behavioral challenges, our proposed intervention centers on promoting joyful engagement and strengthening the parent-child relationship through play.

Parent-child play serves as an ideal context for parents to connect with their young children in a playful and joyful manner, building upon their shared history by incorporating additional positive experiences, thereby supporting their relationship and the children's social and emotional development. Despite the verbal and cognitive barriers children with ID face, play offers them a medium through which they can express their emotions and inner world. Studies on the playfulness of children with ID have found the children's playfulness to be lower than that of TD children [35,36]. Furthermore, early play interventions for children with ID have improved the children's playfulness regardless of the severity of their disability, offering a promising direction given the contributions of play and playfulness to the children's socio-emotional development [37].

Parental playfulness, a relatively understudied characteristic of parenting, is characterized by joyfulness, fun, and creativity and offers parents a unique opportunity to establish a positive communication approach with their children [38]. The only study on parental playfulness among parents of children with ID found them to exhibit lower levels of playfulness compared to parents of TD children. Furthermore, higher levels of parental playfulness were shown to serve as a protective factor against the development of behavior problems in these children [39]. Therefore, early dyadic interventions that focus on fostering playfulness among parents and children with ID may support their relationships as well as advance the children's socio-emotional development, even preventing the development of behavior problems, a major challenge experienced by 30–50% of this population [12,40].

Creative Arts Therapies (CAT), "characterized by the clinical and evidence-informed use of the arts within a therapeutic relationship that relies on experiential and action-based interventions" [41] refer to a variety of methods including musical engagement, visual art, movement and dance, drama/theater, and expressive/creative writing. CAT can provide a fertile terrain for cultivating playfulness and engaging interactions in parents and children with ID, since these therapies are based on non-verbal, creative art-based techniques [41,42]. The unique characteristics of the CAT, the multi-sensorial exploratory experiences, creativity, imagery, and emotional self-expression can help foster interactions between parents and children with ID [41,43]. A recent scoping review of music, dance, and drama therapies for individuals with ID confirmed the positive effects of these therapies on physical, emotional, and cognitive components, as well as on the social skills of this population [44]. However, research on CAT interventions for children with ID and their parents is scarce, although CAT-based dyadic interventions provide significant benefit by fostering playfulness within parent-child interactions in children with ID, thus ameliorating their relationship.

**Study objectives**

This study aims to develop, implement, and assess 'Just Play'(JP) - a dyadic CAT-based intervention that emphasizes the inherent value of dyadic play as an end of itself, encouraging mutual play and enjoyment between children with ID and their mothers. By engaging in spontaneous play, dramatic and musical improvisation, singing, and dyadic movement in a

nonjudgmental and pleasurable space, this study assumes that the dyads will enhance their relationship quality. The trial is a mixed-methods randomized controlled longitudinal study. The study aims to quantitively assess the effectiveness of the JP dyadic intervention to improve mother-child interactions in comparison to psycho-educational parent counseling. The study will examine whether mother and child playfulness serve as mechanisms of change, thereby enhancing the quality of their interactions. The qualitative assessment will explore the mothers' perspective on participating in the intervention, pinpointing its essential characteristics for stimulating maternal playfulness and identifying its long-lasting potential gains on the daily dyadic interactions from the maternal perspective.

### Research questions

- To what extent will the JP dyadic intervention increase the quality of mother-child interactions (as reflected in EA and IPS outcome measures) in comparison to the psycho-educational parent counseling group?
- To what extent will the JP dyadic intervention improve mothers' and children's playfulness in comparison to the psycho-educational parent counseling group?
- How do mothers of children with ID perceive the role of the JP dyadic intervention in shaping their relationships with their children?
- What intervention characteristics and practices do mothers find the most valuable in stimulating and cultivating playful mother-child interactions during the intervention sessions?

### Study hypotheses

- Mothers and children from the JP intervention group will demonstrate greater improvements in mother-child interaction quality as measured by EA and IPS, compared to the psycho-educational parent counseling group.
- Mothers and children from the JP intervention group will show greater improvement in their playfulness outcome measures, compared to the psycho-educational parent counseling group.
- Participation in the JP dyadic intervention will contribute to an improvement in mothers' and children's playfulness, which consequently will mediate the increase in EA and IPS.

## Methods

### Study design

This mixed-methods RCT aims to compare between two conditions: JP dyadic intervention and psycho-educational parent counseling. The quantitative data assessing participants' outcomes from both conditions will be collected at two-time points: before participation in either program and one week upon program completion. The qualitative data will be based on interviews with 12 mothers participating in the JP intervention condition and will be collected at two-time points: one week after participation in the JP intervention and a follow-up two months later. The qualitative component will examine mothers' subjective experiences and meaning-making processes, with particular attention to how they interpret their engagement in the JP dyadic intervention. Incorporating a qualitative arm responds to recent calls for mixed-methods approaches that foreground participants' lived experiences and perspectives, thereby enriching the evaluation of the intervention processes and outcomes [45]. This approach is especially pertinent for dyadic interventions, where relational dynamics and experiential change may not be fully captured by quantitative measures alone. The qualitative strand in this research is intended to provide in-depth understanding of the intervention experience and potential mechanisms underlying its effects, rather than reproducing between-group comparisons.

Fig 1 depicts SPIRIT schedule of enrollment, interventions, and assessments and Fig 2 depicts the study's CONSORT flow diagram [46].

## Participants

The study will include two groups of 30 dyads (total of 60 dyads), each consisting of a mother and her child between the ages of three and seven with a prior diagnosis of ID. The inclusion criteria require that children have a prior diagnosis of ID by a developmental pediatrician. Exclusion criteria include children with an additional primary diagnosis, such as autism spectrum disorder or sensory disorders like blindness or deafness. Participants will be recruited through special education preschools for children with developmental disabilities (which are separate from preschools for children diagnosed with ASD), and through flyers posted in social media groups for parents of children with ID.

| | TRIAL PERIOD | | | | | |
| --- | --- | --- | --- | --- | --- | --- |
| | Enrollment | | Post-randomization | | | |
| TIMEPOINT[b] | $-t_i$ to 0 | 0 | $t_1$ 1st assesment meeting | $t_2$ 8 week intervention period | $t_3$ 2nd assessment meeting | $t_4$ 2 months after completion of intervention |
| **ENROLLMENT:** | | | | | | |
| **Eligibility screen** | X | | | | | |
| **Informed consent** | X | | | | | |
| **Randomization** | | X | | | | |
| **INTERVENTION/ COMPARATOR:** | | | | | | |
| *JP dyadic intervention* | | | | X | | |
| *Parent counseling intervention* | | | | X | | |
| **ASSESSMENTS:** | | | | | | |
| *Descriptive Measures: Demographic questionnaire, VABS 3* | X | | | | | |
| *Outcome Measures: EA, IPS, Playfulness* | | | X | | X | |
| *Qualitative Interviews* | | | | | X | X |

**Fig 1. SPIRIT Schedule of Enrollment, Interventions, and Assessments.**

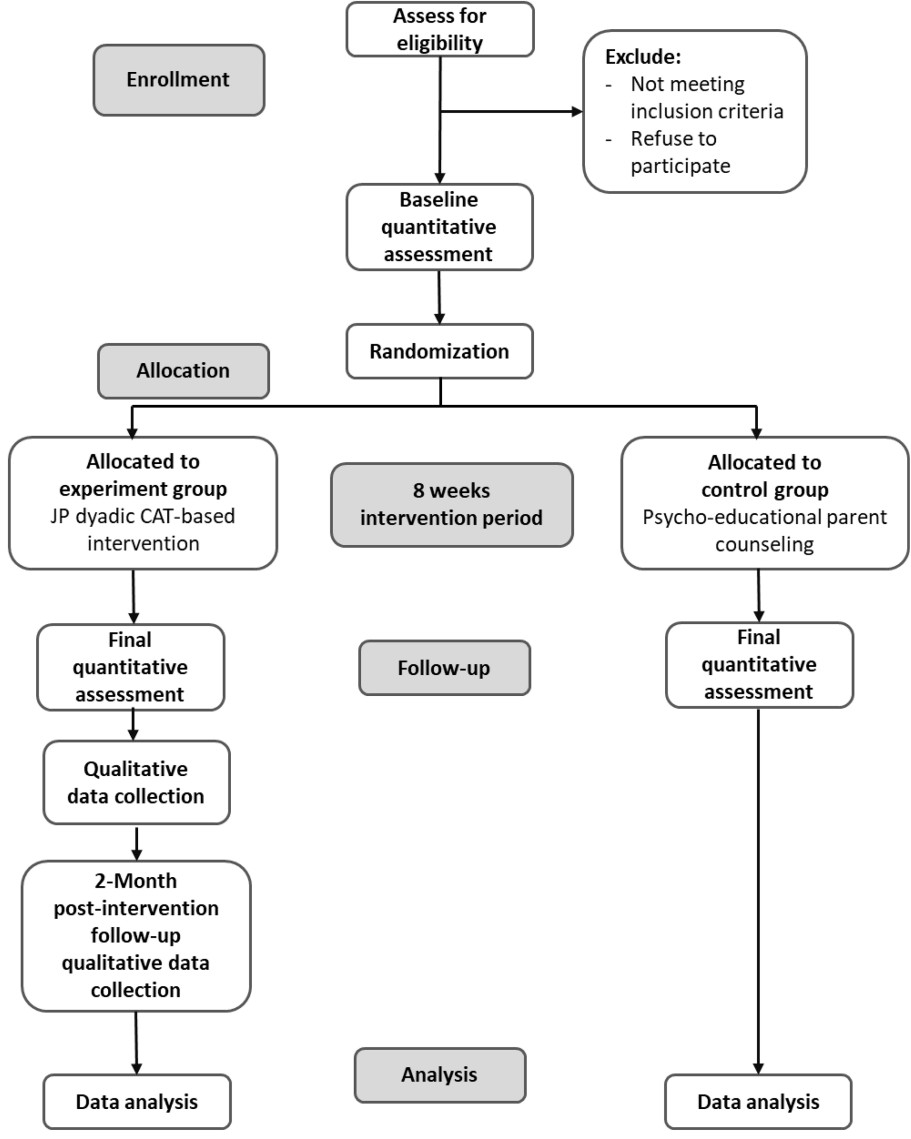

**Fig 2. CONSORT Flow Diagram.**

## Sample size

Previous studies using pre/post EA total sum scores to compare the effectiveness of parent-child interventions [47,48] have reported medium to very large (0.5–1.17) effect size according to Cohen's [49] criteria. The study is powered to detect a large between-group effect size of Cohen's d = 0.8 in the primary outcome of change in EA total sum scores between baseline and post-intervention. With a significance criterion of $\alpha = 0.05$ and power = 80%, the minimum sample size is n = 52 with n = 26 in each sample. Assuming drop-out does not exceed 10%, the aim is to recruit 60 families in total. Furthermore, given the nature of the study involving a short-term home-delivered intervention, n = 30 families in the JP intervention group seems to be a feasible yet significant number.

## Ethics approval and consent

The current study was approved by the ethics committee for the evaluation of research with human subjects of the Faculty of Social Welfare & Health Sciences, University of Haifa, Israel, approval no: 057/23, date of approval: 14 February 2023, and the ethics committee of the Head Scientist of the Israeli Ministry of Education approval no: 1347, date of approval: 10 September 2023. Parents will give written informed consent for their participation and their children's participation in the study and publication of the results. Participant recruitment began February 22nd, 2024 and is currently ongoing.

## Study procedures

After initial contact and providing a comprehensive explanation of the study, the investigator recruiting participants will confirm that eligibility criteria have been met. Following the provision of written informed consent, eligible dyads will be randomly assigned to only one of the conditions – either the JP dyadic intervention or the psycho-educational parent counseling group, using an allocation ratio of 1:1. A permuted block randomization sequence (block size of 4) will be generated using Microsoft Excel. To ensure balance between the conditions, assignments within each block will be randomized using the = RAND() function. The final sequence will be locked into a static spreadsheet to maintain the integrity of the allocation. Participants will be aware of the condition they will participate in, and the study's focus on the quality of the mother-child interaction in the assessment meetings. However, they will not be aware of the study design (that the psycho-educational parent counseling group was chosen as the control group).

After the randomization process, the baseline assessment meeting will be conducted at the participants' homes. The assessment meeting will comprise a mother-child interaction protocol including four consecutive episodes: (a) 10 minutes of free play with a standardized set of toys; (b) 5 minutes of play with a doll and accessories; (c) 7 minutes of social play without toys; and (d) a 3-minute mutual movement activity accompanied by music. All interaction episodes will be video-recorded for subsequent coding. The dyads will then participate in the assigned intervention program. Upon completion of the intervention program, a post-intervention assessment will be conducted using the same mother-child interaction protocol and procedures as the baseline assessment.

In addition, qualitative interviews with a subset of mothers in the JP group will be conducted at post-intervention and two-month follow-up. As the study aims to provide in-depth understanding of the intervention experience and potential mechanisms underlying its effects on the mother-child interactions, the Interpretative Phenomenological Analysis (IPA) methodology was chosen to allow idiographic and phenomenologically grounded exploration of the mothers' experiences. IPA guidelines recommend small, relatively homogeneous samples to permit detailed case-by-case analysis prior to cross-case interpretation [50]. Therefore, interviewing a subsample of 12 mothers from the intervention group will allow for sufficient experiential depth while maintaining analytic rigor consistent with IPA methodology. The mothers will be selected in a purposeful sampling process [50]. All eligible mothers who complete the JP intervention will be invited to take part in a qualitative part of the study immediately following completion of the intervention, and recruitment will continue until the target sample of 12 participants is reached. The interviews will be conducted in the mothers' homes by a member of the research team who was not involved in intervention delivery. The post-intervention interview will be semi-structured and in-depth. The two-month follow-up interview will incorporate an Interpersonal Process Recall procedure, during which the mother and interviewer will jointly watch selected video excerpts from the intervention sessions and reflect on them [51,52]. All interviews will be audio-recorded and transcribed verbatim for qualitative analysis.

## Interventions

This study will compare two conditions: the JP intervention and psycho-educational parent counselling as a control group. Parent counselling, a widely accepted and commonly implemented practice shown to be beneficial for parents of children with ID [53], was selected as the control group for ethical reasons, to ensure participants receive meaningful support

during their participation in the study. It is important to note the differences between these conditions: Psycho-educational parent counselling primarily addresses mothers' experiences of parenting a child with ID, providing guidance, support, and coping strategies. In contrast, the JP intervention specifically targets mother-child play interactions. This difference will also affect how the interventions will be delivered: the JP intervention will involve dyadic play with the mother, child, and therapist in the family's natural home environment, while the psycho-educational parent counseling will involve the mother only and will be conducted online as a preferred method for parent counseling.

### JP dyadic CAT-based intervention

The JP intervention has been developed specifically for this study by the authors of this article, who are trained and experienced creative arts therapists from the fields of music, drama, and visual arts therapies. The JP intervention focuses specifically on the early stages of play: sensorimotor play and object play as these are especially relevant for young children with developmental delays and ID. Given the intervention's goal of improving the quality of mother-child interactions, sensorimotor and object play offer dyads play activities suited to the child's developmental level and offering them opportunities for mutual engagement and enjoyment. By joining the child through familiar play activities, therapist and mother empower the child to lead the interaction, increasing their initiation, motivation, and cooperation. As the intervention progresses, the dyad may even venture into new and more advanced forms of play, such as pretend and social play. The JP intervention aims to enhance the mother-child relationship and improve the child's socioemotional development. The intervention will include eight weekly 45-minute dyadic play sessions with the child, mother, and a music or drama therapist in their home environment. The number of sessions was selected to be eight, since it is considered a minimum 'dosage' for CAT-based family therapy [19]. The intervention will employ and foster musicality, movement, action, and emotional expression, reflecting the cross-modal nature of children's play. The intervention protocol will provide instructions for maintaining intervention fidelity, while allowing flexibility to tailor the meetings to each dyad's strengths and needs.

The intervention's objectives are to: 1. Encourage mothers and children to adopt a playful approach that fosters flexibility, creativity, imagination, and mutual enjoyment. 2. Support and scaffold developmentally appropriate dyadic play interactions based on the child's level and preferences. 3. Promote children's initiation and leading mutual play interactions while enhancing their cooperation skills. 4. Promote sensitive and responsive parental behaviors during play interactions.

The intervention protocol presents a variety of games and ideas according to the different developmental stages of play: sensorimotor play, object play, and pretend play. The therapist will identify existing forms of play and participate in mutual play interactions, while suggesting and introducing novel ideas and games that the dyad may choose to join. The therapist's role as an active player allows them to model playfulness for the child and mother while expanding and developing their play. Play interactions will be advanced by following the child's lead, humor, improvisation, repetition, turn-taking, and exchanging roles. A pilot study was conducted involving six meetings with a mother-child dyad in order to evaluate and refine the intervention protocol. The JP intervention model is depicted in Fig 3. The intervention meetings will be video-recorded for the purpose of supervision, intervention fidelity, and research.

### Psycho-educational parent counseling

The control arm of the psycho-educational parent counseling intervention will consist of eight weekly 45-minute sessions conducted online via Zoom, involving the mother and a trained parent counselor. The sessions will follow the Adlerian model, a well-known and contemporary psycho-educational parent counseling approach supported by ongoing professional training programs [53]. The parent counseling intervention will specifically address key issues and concerns associated with parenting a child with ID such as understanding their developmental needs and challenges. The intervention will provide instructions for maintaining intervention fidelity, while allowing flexibility to tailor the meetings to each mother's strengths and needs in parenting their child with ID.

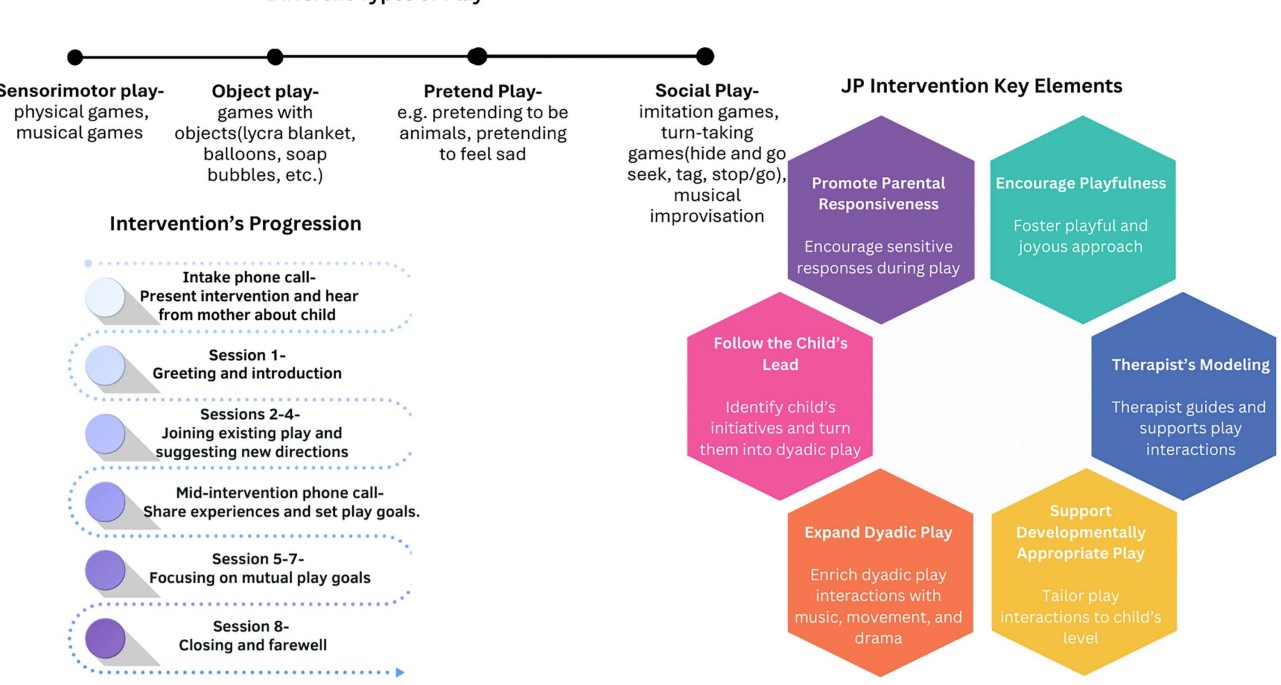

**Fig 3. JP Intervention Model.**

The parent counseling intervention's objectives are: 1. To promote a positive parenting style, by encouragement of the child in order to foster their sense of significance and belonging within their family and the broader social context, as key factors to the children's healthy development. 2. To identify and address maladaptive behaviors in the parent-child relationship, e.g., the parent overprotecting their child with ID, managing challenging child behaviors, or promoting the child's acquisition of new skills. 3. To address maternal stress and family dynamics, e.g., difficulties that may arise between parents, or among siblings of the child with ID [54]. Parent counselors will collaborate with each mother to establish individualized goals to work on during the intervention according to the mother's needs and desires.

## Training interventionists

In the JP intervention condition, music and drama therapists with M.A. certification and previous experience working with children with ID will undergo a three-hour training session on the intervention protocol and will participate in bi-weekly supervision sessions to support their work and ensure treatment fidelity. In the control condition, certified Adlerian parent counsellors will undergo a three-hour training session on counselling parents of children with ID and will participate in bi-weekly supervision sessions by an experienced parent counsellor to support their work and ensure treatment fidelity.

## Assessment of treatment fidelity

To ensure the interventions are delivered as intended by protocol [55], therapists and parent counselors will fill out an implementation checklist after each session and document significant events from the intervention meetings. All dyadic play sessions will be video-recorded to ensure fidelity of the intervention and accurate interpretation of treatment effects [56]. Supervision will be provided by trained and experienced supervisors to maintain method compliance and competence.

## Measures

### Descriptive measures

- A demographic questionnaire will be used to collect background information about the parents' age, ethnicity, education, profession, and income, as well as the participating child's age, diagnosis and treatment, along with additional questions about family members and living situation.

- Vineland Adaptive Behavior Scales-3 parent/caregiver comprehensive form (VABS-3) [57] assesses the child's developmental and adaptive behavior abilities in three core domains: communication, daily living skills, and socialization. The form consists of 381 items scored on a 3-point Likert-type scale ranging from 0 to 2. A higher score on the form indicates higher developmental and adaptive behavior abilities. The VABS-3 comprehensive form will be administered to the child's mother, as it is commonly used to support the diagnosis of intellectual and developmental disabilities. The comprehensive level form, when used with children under the age of seven, has been shown to have excellent internal consistency (Cronbach's α = .99), excellent test-retest reliability of .85, and excellent inter-rater reliability of .86 [57].

### Outcome measures regarding quality of parent-child interaction

#### Primary outcome measure.

- Emotional Availability Scales (EAS) [58] assesses the quality of the parent-child interaction. The coding system is a standardized observational video analysis tool evaluating four dimensions related to parental behaviors: sensitivity, structuring, non-intrusiveness, and non-hostility, and two related to the child's behaviors: responsiveness and involvement. Each dimension is coded on a variety of items scored on a 7-point Likert-type scale with a higher score signifying higher emotional availability. The total sum scores will be calculated for the results. The EAS will be coded based on a 7-minute recording of an episode of social play without toys. The ICC inter-rater reliability has been reported in previous studies to be between .76−.96 [24]. The EAS have been validated for construct in a large number of studies as well as being used in studies with children with ID [21,59] according to the adapted guidelines issued by the EAS authors [60].

#### Secondary outcome measure.

- Parent-Child Interpersonal Synchrony (IPS) will be evaluated by micro-analyzing the movement coordination between the parent and child while moving together to an original musical piece (approximately 3 minutes long) during the two assessment meetings. IPS will be measured using OpenPose, a multi-person 2D pose estimator, to extract skeletal information from the videos. The positions of the mother and child during the mutual movement task will be examined for coherence and variance using the cross-wavelet analysis method, forming an index between 0 and 1 of the degree of interpersonal synchrony between mother and child, with a higher score signifying an optimal outcome of higher interpersonal synchrony, as described in Grinspun et al. [61].

### Mediation measures

- Parental Playfulness Scale (PPS) [62] evaluates the parent's playfulness. The PPS is a standardized observational video analysis tool that assesses parental creativity, imagination, humor, pretend play, and curiosity while playing with their children. The PPS is composed of three subscales: parent's playfulness, following the child's lead, and peer-like behavior rated on a 7-point Likert-type scale from 1 to 7, with a higher score signifying more playfulness. All scores are averaged into a single playfulness score for each parent. The PPS will be scored based on the video-recorded episodes of free play and play with a doll. The PPS has been shown to have high interrater reliability of 0.8 [38,63] and has been used to assess the playfulness of parents to children with ID [27].

- Test of Playfulness (ToP; version 4.2) [64] evaluates the child's playfulness. The ToP is a standardized observational video analysis tool comprised of 30 items rated on a 4-point Likert-type scale from 0 to 3, evaluating the extent, intensity, and skill of the child at play, with a higher score signifying more playfulness. All scores are averaged into a single playfulness score for each child. The ToP will be scored based on the video-recorded episodes of free play and play with a doll. Studies have established the reliability and validity of the ToP determining goodness-of-fit according to the Rasch analysis model when used with children with ID [35,36], and has been used in previous intervention studies [37].

The outcome measures of EA, child playfulness (ToP), and parent playfulness (PPS) will be coded by blind and trained researchers. Interrater reliability will be determined by comparing 20% of the data to an additional blind and trained scorer using the interclass correlation coefficient [65] to establish consistency of independent ratings.

## Data management and storage

Data will be securely stored on an organizational Google Drive account, ensuring the privacy and anonymity of participants. Participants will be identified by code numbers in video recordings and interview data. Personal data will be stored separately with password-protected access. The data will be retained for research purposes for four years and then destroyed.

## Data analyses

### Quantitative data

Univariate statistical analysis will be used to compare demographic and baseline variables between condition groups. To assess the effects of the interventions, a Linear Mixed-Effects Model (LMM) using a Restricted Maximum Likelihood (REML) approach, including time (pre vs. post), group (JP intervention vs. control), and their interaction, will be used. The covariates of the child's gender and the VABS-3 score will be adjusted based on both clinical and statistical relevance. To assess associations between EA and IPS to parent and child playfulness scores, Spearman's correlation coefficients will be calculated. To examine whether the playfulness scores of parents and children (each score entered separately) mediate the relationship between the intervention and the change scores of the primary outcome of EA total sum scores and IPS, Hayes's [66] PROCESS macro will be used, selecting model 4, 5, 7, or 14 based on the relationship between the covariates involved and the study variables. We acknowledge that, using multiple regression with three to four predictors (the treatment group, parents'/child's playfulness score, children's VABS-3 score, and gender), and considering an alpha level of .05, a power of .80, and an effect size of $f^2 = .02$, a sample size of 59–65 participants is required (see Daniel Soper's calculator for a priori sample size [67]). Consequently, given our modest sample size of $n = 60$, the proposed mediation analyses should be regarded as exploratory and interpreted with caution. Moreover, to minimize the risk of overfitting, only covariates with both theoretical and statistical justification (e.g., child's VABS-3 score, gender) will be incorporated into the model, and indirect effects will be estimated via bootstrap confidence intervals.

### Qualitative data

To explore mothers' subjective lived experiences of the JP intervention, an Interpretative Phenomenological Analysis (IPA) methodology will be employed to analyze the data collected from the semi-structured interviews [50]. This approach will facilitate the in-depth exploration of the mothers' experiences during the participation in the JP dyadic intervention through a double hermeneutic – a dual interpretative process in which the researcher will seek to make sense of the mothers' own sense-making [50]. IPA's idiographic orientation will allow detailed examination of each mother's experience while accommodating heterogeneity in how the dyadic intervention was perceived, rather than assuming uniform effects. Interview analysis will follow IPA procedures, progressing from case-level interpretations to patterns of convergence and divergence across participants, with reflexive consideration of the researcher's role, thereby complementing the study's quantitative outcomes.

The two-month follow-up interview will use Interpersonal Process Recall (IPR) procedure, a qualitative, interview-based method designed to access participants' conscious yet unspoken experiences during the original interaction [52]. In this approach, the interaction of interest – here, a dyadic session – will be video-recorded and subsequently reviewed with the mother and the researcher interviewer. The IPR procedure will explore mothers' recollections of their thoughts, feelings, and emotions as they occurred during the recorded session, aiming to reveal internal processes that may not have been previously articulated [68].

## Status and timeline

The study is currently ongoing, beginning in February 2024 and 40 participating dyads have been recruited. 30 have completed the study while the rest are ongoing. Participant recruitment is expected to be completed by June 2026 and full results will be available by September 2026.

## Discussion

The study will discuss the results and clinical applicability of the findings regarding the importance of play and playfulness for the population of children with ID and their mothers. The extensive literature on the importance of play for children's development and the value of playfulness for the parent-child relationship, supports the hypothesis that enhancing play and playfulness of children with ID and their mothers will contribute positively to the quality of their interactions. Therefore, the JP intervention will focus on enhancing parent-child playfulness, while examining its role as a mechanism of change in parent-child interactions. In order to promote playfulness, the JP intervention will utilize CAT and their unique characteristics to facilitate non-verbal, creative, and reciprocal dyadic play. By shifting the focus from behavior management to fostering playfulness and mutual engagement, JP aims to fill an existing gap in early intervention programs for children with ID and their parents.

The study has several limitations: First, although the sample size is powered to detect the expected effects, a larger sample would be recommended in future research. Second, the relatively short duration of the interventions may limit their impact on the outcome measures. Third, participants are not blinded to their assigned condition. Fourth, the inability to fully control for the influence of external factors, such as additional treatments, may confound the results.

Despite these limitations, this study is expected to contribute to the body of research of early interventions for children with ID and their parents, advocating for play-centered approaches and employing CAT. As the application of CAT to support dyadic relationships in early childhood is an emerging area of research, this study is expected to contribute valuable insights to the existing body of knowledge in this field by providing evidence of the effectiveness of dyadic CAT-based interventions in improving mother-child relationships in children with ID. Furthermore, the study findings will shed light on playfulness as a mechanism of change in interactions between children with ID and their mothers, thus informing how CAT can benefit parent-child relationships in this population. The qualitative data will offer a maternal perspective on the intervention's most valuable components and its long-term impact on daily dyadic interactions and parenting experiences. Eventually, the study findings can contribute to the establishment of an evidence-based manual, as well as guidelines for practitioners working with this population.

## Supporting information

**S1 File. Filled SPIRIT Checklist.**
(DOC)

**S2 File. Approved Study Protocol by University of Haifa Ethics Committee.**
(DOCX)

## Acknowledgments

We thank the Emili Sagol Center for Creative Arts Therapies Research Center at the University of Haifa for statistical consultation.

## Author contributions

**Conceptualization:** Rinat Feniger-Schaal, Cochavit Elefant, Limor Goldner, Tal-Chen Rabinowitch.

**Formal analysis:** Limor Goldner.

**Funding acquisition:** Rinat Feniger-Schaal, Cochavit Elefant, Limor Goldner, Tal-Chen Rabinowitch.

**Investigation:** Rita Abramov, Amitai Stern.

**Methodology:** Rinat Feniger-Schaal, Cochavit Elefant, Limor Goldner, Tal-Chen Rabinowitch.

**Writing – original draft:** Rita Abramov, Amitai Stern.

**Writing – review & editing:** Rinat Feniger-Schaal, Cochavit Elefant, Limor Goldner, Tal-Chen Rabinowitch.

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
