## [Decision Letter · Decision Letter 0]

7 Jan 2026

PONE-D-25-45444‘Just Play’ (JP) - creative arts therapies-based dyadic intervention for children with intellectual disability and their mothers: study protocol for a randomized controlled trialPLOS One

Dear Dr. Rita Abramov,

Thank you for submitting your manuscript to PLOS ONE. After careful consideration, we feel that it has merit but does not fully meet PLOS ONE’s publication criteria as it currently stands. Therefore, we invite you to submit a revised version of the manuscript that addresses the points raised during the review process.

We look forward to receiving your revised manuscript.

Kind regards,

Yagnik Dave

Academic Editor

PLOS One

Journal Requirements:

2. PLOS ONE requires that all clinical trials are registered in an appropriate registry (the WHO list of approved registries is at "https://www.who.int/clinical-trials-registry-platform/network/primary-registries" https://www.who.int/clinical-trials-registry-platform/network/primary-registries and more information on trial registration is at http://www.icmje.org/about-icmje/faqs/clinical-trials-registration/).

Please state the name of the registry and the registration number (e.g. ISRCTN or ClinicalTrials.gov) in the submission data and on the title page of your manuscript.

a) Please provide the complete date range for participant recruitment and follow-up in the methods section of your manuscript.

b) If you have not yet registered your trial in an appropriate registry, we now require you to do so and will need confirmation of the trial registry number before we can pass your paper to the next stage of review. Please include in the Methods section of your paper your reasons for not registering this study before enrolment of participants started. Please confirm that all related trials are registered by stating: “The authors confirm that all ongoing and related trials for this drug/intervention are registered”.

Please see http://journals.plos.org/plosone/s/submission-guidelines#loc-clinical-trials for our policies on clinical trials.

3. Thank you for stating in your financial disclosure:

“This study received funding through a competitive peer-reviewed process on the topic of Children with Cognitive Challenges, from an anonymous private donation to the Faculty of Welfare and Health Sciences at the University of Haifa. The external funder had no role in study design, data collection and analysis, decision to publish, or preparation of the manuscript but is updated in yearly reports provided by the investigators. The original grant letter in hebrew and an English translation have been submitted as supplementary files.

AS received Shalem Fund grant no. 890-709 for this study.”

PLOS ONE requires you to include in your manuscript further information about the funder so that any relevant competing interests can be assessed. Please respond to the following questions:

a. Please state whether any of the research costs or authors' salaries were funded, in whole or in part, by a tobacco company (our policy on tobacco funding is at http://journals.plos.org/plosone/s/disclosure-of-funding-sources)

b. Please state whether the donor has any competing interests in relation to this work (see http://journals.plos.org/plosone/s/competing-interests) .

c. Please state whether the identity of the donor might be considered relevant to editors or reviewers’ assessment of the validity of the work.

d. If the donors have no perceived or actual competing interests, please state: “The authors are not aware of any competing interests”.

This information should be included in your cover letter. We will amend your financial disclosure and competing interests on your behalf.

5. We note that you have referenced (ie. Bewick et al. [5]) which has currently not yet been accepted for publication. Please remove this from your References and amend this to state in the body of your manuscript: (ie “Bewick et al. [Unpublished]”) as detailed online in our guide for authors

7. We note that the original protocol that you have uploaded as a Supporting Information file contains an institutional logo. As this logo is likely copyrighted, we ask that you please remove it from this file and upload an updated version upon resubmission.

Additional Editor Comments:

In the current manuscript, major revision is required, especially for the introductory component emphasizing theories of play, the method clarifying randomization, and the discussion focusing on factual limitations.

Reviewers' comments:

Reviewer's Responses to Questions

**Comments to the Author**

1. Does the manuscript provide a valid rationale for the proposed study, with clearly identified and justified research questions?

Reviewer #1: Partly

2. Is the protocol technically sound and planned in a manner that will lead to a meaningful outcome and allow testing the stated hypotheses?

Reviewer #1: Yes

3. Is the methodology feasible and described in sufficient detail to allow the work to be replicable?

Reviewer #1: Yes

4. Have the authors described where all data underlying the findings will be made available when the study is complete?

Reviewer #1: Yes

5. Is the manuscript presented in an intelligible fashion and written in standard English?

Reviewer #1: Yes

6. Review Comments to the Author

You may also provide optional suggestions and comments to authors that they might find helpful in planning their study.

Reviewer #1: Thank you for the opportunity to review this interesting work. I found the protocol to be well-written; however, there are several points that the authors may want to elaborate on.

Introduction:

- The foundation of the study is the concept of play in developmental stages. More descriptions are needed. Key theorists should be mentioned to provide the readers with a clearer understanding of play. Additionally, more information is needed regarding the importance of play and its impact on children with intellectual disabilities (ID) and their mothers. What are the potential drawbacks if the ability to play is not adequately developed in children with ID?

- Could the authors explain why a psychoeducational parent counselling group was selected as the comparison group?

Methods:

- Only the participants in the intervention group will undergo a 2-month follow-up assessment, whereas the participants in the control group will not be required to do so. This procedure is somewhat unusual. For a more robust design, both groups should be assessed at the same time point. Justification for this design is necessary. Additionally, will the participants in the control group have the opportunity to participate in the JP intervention as part of a waitlist control design?

- While additional information has been provided regarding the design of the intervention, more details are needed, particularly in how the key elements of the JP intervention relate to the stages of play, and how these stages of play are connected to progress within the intervention.

- More descriptions are needed regarding the randomisation and blinding procedures, such as who will be responsible for conducting these procedures and how the computer program operates.

- The sample size for the quantitative data analysis is well justified; however, a justification is needed for the sample size used in conducting the qualitative interviews. Furthermore, how will this subsample be recruited?

- Additional descriptions are necessary regarding the procedures for conducting qualitative interviews and the data analysis.

- Where will the participants be recruited from? Since children with autism spectrum disorder (ASD) will be excluded from the study, how can the authors ensure that they will recruit a sufficient number of participants within the study period, given the high comorbidity between ASD and ID?

Discussion:

- The limitations section should include the fact that the participants are not blinded to the group condition.

7. PLOS authors have the option to publish the peer review history of their article (what does this mean?). If published, this will include your full peer review and any attached files.

Reviewer #1: No

---

## [Author Response · Author response to Decision Letter 1]

20 Feb 2026

Dear Editor,

We thank you and the reviewer for your careful review and constructive comments on our manuscript. We have addressed all points raised and revised the manuscript accordingly. We believe these revisions have improved the clarity and rigor of the paper and hope that the revised version is now suitable for publication in PLOS ONE.

Below, we provide a point-by-point response to each comment. All revisions in the manuscript have been marked using track changes.

Sincerely,

Rita Abramov

on behalf of all authors

Editor comments

Response: The manuscript has been revised in accordance with PLOS ONE’s style requirements.

2. PLOS ONE requires that all clinical trials are registered in an appropriate registry.

Please state the name of the registry and the registration number (e.g. ISRCTN or ClinicalTrials.gov) in the submission data and on the title page of your manuscript.

a) Please provide the complete date range for participant recruitment and follow-up in the methods section of your manuscript.

The name of the trial registry and the corresponding registration number have been added to the title page of the manuscript (ln 19-20). In addition, the Methods section has been revised to include the complete date range for participant recruitment and follow-up (ln 276-277).

3. PLOS ONE requires you to include in your manuscript further information about the funder so that any relevant competing interests can be assessed. This information should be included in your cover letter. We will amend your financial disclosure and competing interests on your behalf.

Response: The updated cover letter includes the following information:

“The authors declare that they have no competing interests related to the funder. The funder had no role in study design, data collection and analysis, decision to publish, or preparation of the manuscript.”

Response:The updated submission form includes the following Data Availability Statement:

“All data regarding the study protocol are in the manuscript and supporting information files.”

5. We note that you have referenced (ie. Bewick et al. [5]) which has currently not yet been accepted for publication.

Response: We were unable to identify the referenced citation in the current version of the manuscript.

6. Please include captions for your Supporting Information files at the end of your manuscript, and update any in-text citations to match accordingly

Response: The manuscript has been revised to include captions for all Supporting Information files at the end of the document (ln 708-713).

7. We note that the original protocol that you have uploaded as a Supporting Information file contains an institutional logo. As this logo is likely copyrighted, we ask that you please remove it from this file and upload an updated version upon resubmission.

Response: The institutional logos have been removed from S2_File, S3_File, S4_File, and S5_Files and updated versions have been uploaded accordingly.

8. In the current manuscript, major revision is required, especially for the introductory component emphasizing theories of play, the method clarifying randomization, and the discussion focusing on factual limitations.

Response: We appreciate the Editor’s constructive comments. The manuscript has been thoroughly revised in response. The introduction now provides a more detailed theoretical framing of play (ln 67-108), the methods section clarifies the randomization procedure (ln 283-286), and the discussion more explicitly addresses the study’s limitations (ln 512-517). We believe these revisions have substantially strengthened the manuscript.

Reviewer comments

1. Introduction: The foundation of the study is the concept of play in developmental stages. More descriptions are needed. Key theorists should be mentioned to provide the readers with a clearer understanding of play. Additionally, more information is needed regarding the importance of play and its impact on children with intellectual disabilities (ID) and their mothers. What are the potential drawbacks if the ability to play is not adequately developed in children with ID?

Response: Thank you for this important comment. The introduction has been revised to expand the theoretical and developmental understating of play, highlight key theorists, and discuss its significance for children with ID and their mothers (ln 67-108).

2. Could the authors explain why a psychoeducational parent counselling group was selected as the comparison group?

Response: We have further elaborated on the selection of the psycho-educational parent counselling group as the comparison condition (ln 318-328).

3. Methods: Only the participants in the intervention group will undergo a 2-month follow-up assessment, whereas the participants in the control group will not be required to do so. This procedure is somewhat unusual. For a more robust design, both groups should be assessed at the same time point. Justification for this design is necessary.

Response: A justification for conducting the 2-month follow-up assessment only in the intervention group has been added to the manuscript (ln 237-246).

4. Will the participants in the control group have the opportunity to participate in the JP intervention as part of a waitlist control design?

Response: As the present study does not employ a waitlist control design, dyads allocated to the psycho-educational parent counselling condition will not be offered subsequent participation in the JP intervention within the framework of this trial. Furthermore, clarification on the study procedures has been added to the manuscript (ln 311-314).

5. While additional information has been provided regarding the design of the intervention, more details are needed, particularly in how the key elements of the JP intervention relate to the stages of play, and how these stages of play are connected to progress within the intervention.

Response: The manuscript has been revised to provide additional details on how the key elements of the JP intervention correspond to the stages of play and how these stages relate to progress within the intervention (ln 332-341).

6. More descriptions are needed regarding the randomization and blinding procedures, such as who will be responsible for conducting these procedures and how the computer program operates.

Response: The manuscript has been revised to provide expanded and clarified descriptions of both the randomization and blinding procedures (ln 283-289).

7. The sample size for the quantitative data analysis is well justified; however, a justification is needed for the sample size used in conducting the qualitative interviews. Furthermore, how will this subsample be recruited?

Response: The manuscript has been revised to provide a more detailed description of the qualitative procedures, including the rationale for the sample size (ln 298-315).

8. Additional descriptions are necessary regarding the procedures for conducting qualitative interviews and the data analysis.

Response: A Qualitative Data Analysis section has been added to describe the procedures for analyzing the qualitative study data (ln 476-494).

9. Where will the participants be recruited from? Since children with autism spectrum disorder (ASD) will be excluded from the study, how can the authors ensure that they will recruit a sufficient number of participants within the study period, given the high comorbidity between ASD and ID?

Response: Thank you for this comment regarding the current clinical reality. The manuscript has been revised to clarify the participant recruitment procedures (ln 257-258).

10. Discussion: The limitations section should include the fact that the participants are not blinded to the group condition.

Response: The manuscript has been revised to explicitly address the lack of participant blinding in addition to other limitation (ln 512-517).

---

## [Decision Letter · Decision Letter 1]

16 Mar 2026

PONE-D-25-45444R1‘Just Play’ (JP) - creative arts therapies-based dyadic intervention for children with intellectual disability and their mothers: study protocol for a mixed-methods randomized controlled trialPLOS One

Dear Dr. Abramov,

Thank you for submitting your manuscript to PLOS ONE. After careful consideration, we feel that it has merit but does not fully meet PLOS ONE’s publication criteria as it currently stands. Therefore, we invite you to submit a revised version of the manuscript that addresses the points raised during the review process.

(a) The primary analysis plan is too simplistic for the number and structure of outcomes. A two-way repeated-measures ANOVA for EA and IPS only, with Time = pre/post and Condition = intervention/control, was proposed. However, it's not clear, if each EA dimension (EA has multiple subscales) will be analyzed separately. Will multiplicity be controlled. Also, it's not clear, what would be the "primary outcome".

(b) Again, the sample size is based on change in EA scales, but it's not clear (to me) that the EA is the primary outcome variable.

(c) The analytical method - 2-way repeated measures ANOVA, maynot be adequate in presence of dropouts/missing data. Alternative linear mixed models, with a missing data imputation technique embedded, appears more elegant. On the overall, the methodology for handling missing data is not clearly written.

(d) The mediation analysis plan appeared too optimistic, mostly due to the small sample size issue in this kind of small randomized trials, as well as modeling simple change scores.

(e) In randomized trials, covariate adjustment should be based on clinical relevance or prespecification, not on baseline significance tests, which are notoriously unhelpful and sample-size dependent; read here: https://pubmed.ncbi.nlm.nih.gov/12325108/

Hence, comments like: "If any demographic variable differs significantly between groups, it will be entered as a covariate in the ANOVA models." are misleading.

We look forward to receiving your revised manuscript.

Kind regards,

Yagnik Dave

Academic Editor

PLOS One

Journal Requirements:

Reviewers' comments:

Reviewer's Responses to Questions

(a) The primary analysis plan is too simplistic for the number and structure of outcomes. A two-way repeated-measures ANOVA for EA and IPS only, with Time = pre/post and Condition = intervention/control, was proposed. However, it's not clear, if each EA dimension (EA has multiple subscales) will be analyzed separately. Will multiplicity be controlled. Also, it's not clear, what would be the "primary outcome".

(b) Again, the sample size is based on change in EA scales, but it's not clear (to me) that the EA is the primary outcome variable.

(c) The analytical method - 2-way repeated measures ANOVA, maynot be adequate in presence of dropouts/missing data. Alternative linear mixed models, with a missing data imputation technique embedded, appears more elegant. On the overall, the methodology for handling missing data is not clearly written.

(d) The mediation analysis plan appeared too optimistic, mostly due to the small sample size issue in this kind of small randomized trials, as well as modeling simple change scores.

(e) In randomized trials, covariate adjustment should be based on clinical relevance or prespecification, not on baseline significance tests, which are notoriously unhelpful and sample-size dependent; read here: https://pubmed.ncbi.nlm.nih.gov/12325108/

Hence, comments like: "If any demographic variable differs significantly between groups, it will be entered as a covariate in the ANOVA models." are misleading.

**Comments to the Author**

1. Does the manuscript provide a valid rationale for the proposed study, with clearly identified and justified research questions?

Reviewer #2: Partly

2. Is the protocol technically sound and planned in a manner that will lead to a meaningful outcome and allow testing the stated hypotheses?

Reviewer #2: Partly

3. Is the methodology feasible and described in sufficient detail to allow the work to be replicable?

Reviewer #2: No

4. Have the authors described where all data underlying the findings will be made available when the study is complete?

Reviewer #2: Yes

5. Is the manuscript presented in an intelligible fashion and written in standard English?

Reviewer #2: Yes

6. Review Comments to the Author

You may also provide optional suggestions and comments to authors that they might find helpful in planning their study.

Reviewer #2: The authors were able to address the previous rounds of comments with satisfaction. However, I have additional questions:

(a) The primary analysis plan is too simplistic for the number and structure of outcomes. A two-way repeated-measures ANOVA for EA and IPS only, with Time = pre/post and Condition = intervention/control, was proposed. However, it's not clear, if each EA dimension (EA has multiple subscales) will be analyzed separately. Will multiplicity be controlled. Also, it's not clear, what would be the "primary outcome".

(b) Again, the sample size is based on change in EA scales, but it's not clear (to me) that the EA is the primary outcome variable.

(c) The analytical method - 2-way repeated measures ANOVA, maynot be adequate in presence of dropouts/missing data. Alternative linear mixed models, with a missing data imputation technique embedded, appears more elegant. On the overall, the methodology for handling missing data is not clearly written.

(d) The mediation analysis plan appeared too optimistic, mostly due to the small sample size issue in this kind of small randomized trials, as well as modeling simple change scores.

(e) In randomized trials, covariate adjustment should be based on clinical relevance or prespecification, not on baseline significance tests, which are notoriously unhelpful and sample-size dependent; read here: https://pubmed.ncbi.nlm.nih.gov/12325108/

Hence, comments like: "If any demographic variable differs significantly between groups, it will be entered as a covariate in the ANOVA models." are misleading.

7. PLOS authors have the option to publish the peer review history of their article (what does this mean?). If published, this will include your full peer review and any attached files.

Reviewer #2: No

---

## [Author Response · Author response to Decision Letter 2]

16 Apr 2026

a) The primary analysis plan is too simplistic for the number and structure of outcomes. A two-way repeated-measures ANOVA for EA and IPS only, with Time = pre/post and Condition = intervention/control, was proposed. However, it's not clear, if each EA dimension (EA has multiple subscales) will be analyzed separately. Will multiplicity be controlled. Also, it's not clear, what would be the "primary outcome".

(b) Again, the sample size is based on change in EA scales, but it's not clear (to me) that the EA is the primary outcome variable.

Response: Thank you for these comments. We have now explicitly designated the EA total sum score as the primary outcome for this study, as well as changing the analytical method, thereby reducing the risk of multiplicity (ln 466-471).

(c) The analytical method - 2-way repeated measures ANOVA, may not be adequate in presence of dropouts/missing data. Alternative linear mixed models, with a missing data imputation technique embedded, appears more elegant. On the overall, the methodology for handling missing data is not clearly written.

Response: To address dropout and potential missing data, we will use Linear Mixed-Effects Models (LMM) using a Restricted Maximum Likelihood (REML) approach, which includes time, group, and their interaction effects. Given that all effects were tested within a single pre-specified model, no correction for multiplicity will be applied, consistent with recommendations in the statistical literature (e.g., Rothman, 1990; Fitzmaurice et al., 2011) (ln 466-471).

(d) The mediation analysis plan appeared too optimistic, mostly due to the small sample size issue in this kind of small randomized trials, as well as modeling simple change scores.

Response: We have addressed this important comment and revised the mediation analysis plan. Given our modest sample size (n=60), the mediation analyses should be considered exploratory and interpreted with caution. To reduce the risk of overfitting, only predictors with theoretical (treatment group, child’s VABS-3 score) and statistical justification will be included in the model, and indirect effects will be estimated using bootstrap confidence intervals (ln 471-483).

(e) In randomized trials, covariate adjustment should be based on clinical relevance or pre-specification, not on baseline significance tests, which are notoriously unhelpful and sample-size dependent; read here: https://pubmed.ncbi.nlm.nih.gov/12325108/

Hence, comments like: "If any demographic variable differs significantly between groups, it will be entered as a covariate in the ANOVA models." are misleading.

Response: We thank the reviewer for this important comment. We have changed the analysis so the covariates will be adjusted based on clinical relevance: the child’s gender and their VABS-3 score (ln 480-483).

---

## [Decision Letter · Decision Letter 2]

3 May 2026

‘Just Play’ (JP) - creative arts therapies-based dyadic intervention for children with intellectual disability and their mothers: study protocol for a mixed-methods randomized controlled trial

PONE-D-25-45444R2

Dear Dr. Rita Abramov

We’re pleased to inform you that your manuscript has been judged scientifically suitable for publication and will be formally accepted for publication once it meets all outstanding technical requirements.

Kind regards,

Yagnik Dave

Academic Editor

PLOS One

Additional Editor Comments (optional):

Reviewers' comments:

Reviewer's Responses to Questions

**Comments to the Author**

1. Does the manuscript provide a valid rationale for the proposed study, with clearly identified and justified research questions?

Reviewer #2: Yes

2. Is the protocol technically sound and planned in a manner that will lead to a meaningful outcome and allow testing the stated hypotheses?

Reviewer #2: Yes

3. Is the methodology feasible and described in sufficient detail to allow the work to be replicable?

Reviewer #2: Yes

4. Have the authors described where all data underlying the findings will be made available when the study is complete?

Reviewer #2: Yes

5. Is the manuscript presented in an intelligible fashion and written in standard English?

Reviewer #2: Yes

6. Review Comments to the Author

You may also provide optional suggestions and comments to authors that they might find helpful in planning their study.

Reviewer #2: The authors addressed my previous round of comments with satisfaction. I have no further comments this round.

7. PLOS authors have the option to publish the peer review history of their article (what does this mean?). If published, this will include your full peer review and any attached files.

Reviewer #2: No

---

## [Editor Report · Acceptance letter]

PONE-D-25-45444R2

PLOS One

Dear Dr. Abramov,

I'm pleased to inform you that your manuscript has been deemed suitable for publication in PLOS One. Congratulations! Your manuscript is now being handed over to our production team.

Kind regards,

on behalf of

Dr. Yagnik Dave

Academic Editor

PLOS One